# PLA2G12A as a Novel Biomarker for Colorectal Cancer with Prognostic Relevance

**DOI:** 10.3390/ijms241310889

**Published:** 2023-06-30

**Authors:** Eva Parisi, Ivan Hidalgo, Robert Montal, Ona Pallisé, Jordi Tarragona, Anabel Sorolla, Anna Novell, Kyra Campbell, Maria Alba Sorolla, Andreu Casali, Antonieta Salud

**Affiliations:** 1Research Group of Cancer Biomarkers, Biomedical Research Institute of Lleida (IRBLleida), 25198 Lleida, Spain; 2Department of Experimental Medicine, University of Lleida, 25198 Lleida, Spain; 3Department of Medical Oncology, Arnau de Vilanova University Hospital (HUAV), 25198 Lleida, Spain; 4Department of Basic Medical Sciences, University of Lleida and IRBLleida, 25198 Lleida, Spain; 5Department of Pathology and Molecular Genetics/Oncologic Pathology Group, Biomedical Research Institute of Lleida (IRBLleida), University of Lleida, CIBERONC, 25198 Lleida, Spain; 6School of Biosciences, The University of Sheffield, Sheffield S10 2TN, UK; 7Department of Medicine, University of Lleida, 25198 Lleida, Spain

**Keywords:** PLA2G12A, colorectal cancer, prognosis, *Drosophila*

## Abstract

Metastasis is the leading cause of colorectal cancer (CRC)-related deaths. Therefore, the identification of accurate biomarkers predictive of metastasis is needed to better stratify high-risk patients to provide preferred management and reduce mortality. In this study, we identified 13 new genes that modified circulating tumor cell numbers using a genome-wide genetic screen in a whole animal CRC model. Candidate genes were subsequently evaluated at the gene expression level in both an internal human CRC cohort of 153 patients and an independent cohort from the TCGA including 592 patients. Interestingly, the expression of one candidate, *PLA2G12A*, significantly correlated with both the time to recurrence and overall survival in our CRC cohort, with its low expression being an indicator of a poor clinical outcome. By examining the TCGA cohort, we also found that low expression of *PLA2G12A* was significantly enriched in epithelial–mesenchymal transition signatures. Finally, the candidate functionality was validated in vitro using three different colon cancer cell lines, revealing that *PLA2G12A* deficiency increases cell proliferation, migration, and invasion. Overall, our study identifies PLA2G12A as a prognostic biomarker of early-stage CRC, providing evidence that its deficiency promotes tumor growth and dissemination.

## 1. Introduction

Colorectal cancer (CRC) is one of the most frequently diagnosed cancers worldwide [1]. However, its incidence has slightly decreased from 1976 to 2005 in the US [2] and the same happened with mortality, now being more than 50% lower than the maximum mortality rates [3]. These improvements are due to the implementation of cancer prevention and screening programs, as well as better treatment modalities. Conversely, the CRC incidence and mortality shows age-dependent trends, being specially increased in people aged <50 [4].

It is estimated that 40–50% of CRC patients will present with metastases throughout the disease [5]. Indeed, metastasis is the main cause of cancer-related deaths; thus, a better understanding of this multistep process is urgently needed. In particular, additional tools that could help to better stratify patients based on the recurrence risk would help clinicians in decision making and thus optimize current treatments.

Regarding CRC treatment, adjuvant chemotherapy with fluoropyrimidine and oxaliplatin is recommended after a surgical excision for stages III and II at a high risk [6]. A risk assessment is based on clinic-pathological factors such as the number of lymph nodes analyzed after surgery (less than 12), poorly differentiated histology, lymphatic/vascular invasion, perineural invasion, tumor budding, bowel obstruction, localized perforation, positive margins, and stable microsatellite status (MSS) [7]. Among these factors, lymph node sampling (less than 12) and pT4 are currently recognized as the major prognostic parameters associated with worse survival, regardless of MS [8]. According to the National Comprehensive Cancer Network guidelines, stage II MSS patients categorized as low-risk are recommended to be only under observation or submitted to not oxaliplatin adjuvant chemotherapy (MOSAIC trial [9]). Nevertheless, the question is whether or not the treatment has to be given. In addition, in high-risk MSS stage II and all stage III patients, the recommendations are the administration of fluoropyrimidines plus oxaliplatin (FOLFOX or CAPEOX regimens) for a variable time [10,11]. However, there are no data correlating risk characteristics and a chemotherapy regimen selection.

Several biomarkers have also been proposed to improve recurrence prediction, such as CDX2, a nuclear protein essential for the proliferation and development of intestinal epithelial cells and that is frequently downregulated during tumorigenesis [12]. Dalerba et al. indicated that a lack of CDX2 could identify a subgroup of high-risk stage II CRC patients who could benefit from adjuvant chemotherapy [13]. Another study showed that the presence of circulating tumor DNA (ctDNA) after chemotherapy is associated with a lower recurrence-free survival (RFS) in stage II patients compared with those not having detectable ctDNA [14]. Furthermore, gene expression profiles have been analyzed to identify correlations with the clinical course of CRC [15]. Oncotype DX colon cancer [16], ColoPrint [17], and ColDx [18] assays performed well in predicting recurrence in stage II/III patients independently of other classical risk factors. Nevertheless, the international guidelines did not recommend the use of these panels, as they failed to predict the benefit of adjuvant chemotherapy; therefore, further validation is required.

During cancer progression, carcinoma cells often undergo epithelial–mesenchymal transition (EMT), where they acquire enhanced migratory abilities that enable them to disseminate and form secondary metastases at distant sites [19]. Preclinical in vivo models of metastasis are needed to obtain novel insights with potential clinical implications. Several organisms have been used to model metastasis in vivo, mainly mice and rats [20]. Other inferior organisms have also been employed to model metastasis or at least, to decipher the events preceding it in zebrafish [21], in worms [22], and also in yeast [23]. In addition, in vivo fly models have been useful in addressing the roles of several signaling pathways implicated in mammalian tumorigenesis and metastasis [24,25]. Specifically, we used a model of intestinal tumors in Drosophila melanogaster based on the activation of the Wnt and EFGR/Ras pathways, which reproduced many features of human CRC [26,27].

In this study, we used this innovative *Drosophila melanogaster* CRC model to screen several genes, searching for candidates that modify the migration capabilities of tumor cells. These results were evaluated at the gene expression level in human CRC cohorts and were functionally validated in vitro. Interestingly, our results reveal that *PLA2G12A* can act as a tumor suppressor gene being involved in dissemination and associated with a good prognosis of CRC.

## 2. Results

### 2.1. Screening of Candidate Genes Involved in Tumor Cell Dissemination Using an In Vivo CRC Model

We used a non-metastatic CRC *Drosophila* model to search for candidate genes with a functional role in primary tumor cell dissemination. We selected 97 genes downregulated in a *Drosophila* metastatic CRC model from previous experiments [27] that may play a role in inhibiting cell dissemination in vivo. To test this hypothesis, we expressed an RNAi transgene for each of these genes in primary non-metastatic tumor cells that also expressed luciferase, the activity of which correlated linearly with the number of tumor cells present [27]. Through a highly sensitive luciferase assay, we quantified the number of circulating tumor cells (CTCs) in the hemolymph (analogous to the blood in vertebrates) and the primary tumor burden (Appendix A). We identified 13 genes that showed a statistically significant increase in the number of CTCs, tumor cells that had broken through the basement surrounding the primary tumor and disseminated into the hemolymph, compared to control flies bearing non-metastatic tumors (Figure 1A) (Appendix A). We conclude that these 13 genes may act as potential tumor suppressor genes because their inhibition in the fly intestine increases the number of CTCs. For technical reasons, we continued our study with 10 of these genes.

### 2.2. Identification of PLA2G12A as a Potential Tumor Suppressor Gene Associated with Prognosis in Human CRC Cohorts

To evaluate the clinical relevance of the *Drosophila* functional screening, we analyzed the impact of candidate genes on TTR and OS in our internal CRC human HUAV cohort (*n* = 153). First, we determined the human orthologues of the *Drosophila* genes and analyzed RNA gene expression for the 10 candidates. Among all genes (Table 1), the expression of *PLA2G12A* was the only one with a significant association with both the time to recurrence (TTR) (HR = 0.590, 95% CI: 0.375–0.928, *p* = 0.022) (Figure 2A) and overall survival (OS) (HR = 0.433, 95% CI: 0.190–0.989, *p* = 0.047) (Figure 2B), where its low expression indicated a poor clinical outcome in terms of a higher risk of recurrence and subsequent death.

Interestingly, this higher risk of recurrence determined by *PLA2G12A* expression was maintained in stage II tumors (HR = 0.546, 95% CI: 0.308–0.967, *p* = 0.038) (Figure 2C), the ones with less evidence of a benefit from adjuvant treatments. Notably, no other clinico-pathological features were associated with *PLA2G12A* expression (Table 2), indicating that its biological relevance is independent of other classic prognostic factors. *PLA2G12A* codify for a member of the family of secreted phospholipases A2 (PLA2s), enzymes that hydrolyze phospholipids and are involved in several processes related to inflammation and cancer [28,29].

To further validate the prognostic properties of the candidate genes, we used the external CRC from The Cancer Genome Atlas (TCGA) cohort (*n* = 592) with available transcriptome data (Appendix A). Again, a low expression of *PLA2G12A* was significantly associated with a higher risk of recurrence or death in terms of the composite endpoint recurrence-free survival (RFS) (HR = 0.580, 95% CI: 0.414–0.813, *p* = 0.002) (Figure 2D). Finally, to explore the biological underpinnings of PLA2G12A in CRC, we analyzed 50 hallmark gene expression signatures with a gene set enrichment analysis (GSEA) in the TCGA cohort (Appendix A, Figure 2E). The analysis revealed that a low *PLA2G12A* expression was significantly enriched in EMT signatures (ES = 0.64, *p* = 0.012) (Figure 2F), reinforcing its potential role in the first steps of the metastatic process.

### 2.3. Functional Characterization of PLA2G12A in CRC Preclinical Models

To further investigate the potential of PLA2G12A as a clinical prognostic marker, we assessed its role in tumor growth and development. In the *Drosophila* CRC model previously described, the inhibition of GXIVsPLA2 (the fly orthologue of *PLA2G12A*) not only increased the number of CTCs but also increased the intestinal primary tumor burden (Appendix A, Figure 1D). As these primary tumor cells also express GFP, we dissected several midguts and measured the total tumor area, as described previously by Adams et al. [30]. Our results showed that the inhibition of GXIVsPLA2 induced the spread of primary tumors along the midgut compared to midguts bearing control Apc-Ras tumors (Figure 1B,C). Together, our results suggest a role for GXIVsPLA2 in inhibiting both intestinal primary tumor growth and the invasion of tumor cells into the hemolymph.

We also performed in vitro experiments with CC cell lines to investigate possible changes in cell behavior induced by *PLA2G12A* expression. To this end, we used three CC cell lines (HCT116, HT29, and SW480) with distinct proliferation and migration capacities. As shown in Figure 3A, *PLA2G12A* inhibition induces an increase in proliferation rates compared with the control in all three cell lines. HCT116 and HT29 cells showed proliferation increments 48 h after transfection, whereas for SW480 cells, the effect appeared to be significant 72 h post-transfection. We next analyzed migration rate changes caused by *PLA2G12A* inhibition using a scratch wound healing assay, where transfected cells were seeded and the migratory ability of the cells at covering the scratch was monitored. Figure 3B shows three different time points in the experiment and representative images of the migration rate measurement. According to our observations, *PLA2G12A*-downregulated cells showed increased migration rates compared to control cells, as significant changes were seen in both HCT116 and HT29 cell lines.

Lastly, we analyzed whether *PLA2G12A* expression could alter other cell aspects related to the metastatic process, such as cell invasion and anchorage-independent cell growth abilities. To study these processes, we transfected and seeded cells in a transwell chamber treated with matrigel and incubated them for 48 h. We noticed that *PLA2G12A*-inhibited cells appeared in a greater number in the bottom layer of the chamber than in the control, indicating an increased invasion capacity when *PLA2G12A* levels were low in the three CC cell lines used (Figure 4A). By analyzing the growth of transfected cells in an agar layer, we found that *PLA2G12A*-inhibited cells formed a more significant number of colonies than the control cells (Figure 4B) in all cell lines studied. Together, these results provide evidence that *PLA2G12A* inhibition promotes the cell proliferation, migration, invasion, and anchorage-independent growth of CC cells.

## 3. Discussion

In this study, we performed wide molecular screening of genes potentially involved in tumor progression in an in vivo *Drosophila* CRC model and a subsequent validation of its prognostic power in two independent human cohorts, and discovered a novel putative tumor suppressor gene for CRC, *PLA2G12A*. This gene, a phospholipase A2 family member, can modify tumor growth and dissemination in *Drosophila*, suggesting that it could be a potential driving factor for CRC metastatic tumorigenicity.

Despite numerous efforts, it has been difficult to find a reliable recurrence risk biomarker for CRC that could select patients requiring adjuvant therapies. In this regard, some gene expression panels have been used to add value in predicting a prognosis in stage II/III patients [16,17,18]. In fact, several ongoing clinical trials have aimed to validate these gene expression signatures for a recurrence risk assessment in these patients. For instance, the PARSC trial (NCT00903565) focused on assessing the utility of the ColoPrint assay to estimate the 3-year relapse rate in stage II CRC patients.

In this work, we focused our research on searching for candidates that directly modify tumor behavior in an in vivo *Drosophila* model. The fruit fly *Drosophila melanogaster* has become an important model system for cancer studies because of its potential for conducting large-scale genetic screens [31]. Moreover, research on the signaling pathways involved in tumor formation, such as Hippo, Notch, Dpp, and JAK-STAT, has been enriched with investigations performed in *Drosophila* [32,33,34]. Our model is based on the induction of mutant clones in the Wnt pathway (Apc and Apc2) in the adult fly intestine, and the overexpression of the oncogenic form of Ras, Ras^V12^, facilitating the generation of tumor-like overgrowths similar to those found in human CRC tumors. These are characterized by an increased proliferation, blockage of cell differentiation, alterations in cell polarity, and disruption of the organ architecture [26]. No sign of cell dissemination was found in these Apc-Ras flies, whose clones were confined to the gut and surrounded by a thick layer of the basement membrane [26]. For these reasons, the model is an excellent tool to easily detect molecules able to promote tumor cell migration and propagation outside the gut. The molecular screening of 97 different candidates identified 13 genes that increased in vivo tumor cell dissemination. Each of these genes is capable of changing the number of CTCs that can escape from the primary tumor and be detected in the hemolymph, which is analogous to human blood.

To determine whether the candidates obtained in the *Drosophila* model maintain invasive capabilities in more complex organisms, we analyzed their orthologous genetic expression in two independent cohorts of CRC patients. We found one gene, *PLA2G12A*, whose low expression was correlated with tumor recurrence and a poor clinical outcome. PLA2G12A is a member of the family of phospholipases A2 (PLA2s), a large superfamily of enzymes that hydrolyzes phospholipids and releases fatty acids and lysophospholipids [35].

Phospholipases can be divided into three major classes—PLA (consisting of A1 and A2), PLC, and PLD, which are differentiated by the type of reaction that they catalyze [36]. Moreover, each class of phospholipase is composed of many isotypes with distinct functions, domains, and regulatory mechanisms [37,38,39]. Within the phospholipase A2 (PLA2) superfamily, secreted PLA2 (sPLA2) enzymes comprise the largest family containing 12 mammalian isoforms with a conserved catalytic site. Every group of sPLA2s exhibit unique tissue and cellular localizations with specific enzymatic properties, suggesting distinct biological roles [40]. In humans, two PLA2G12, PLA2G12A, and a catalytically inactive PLA2G12A-like protein (PLA2G12B) were cloned [41,42]. The functional characterization of PLA2G12A revealed that its catalytic activity is relatively low in comparison with sPLA2s. While the enzymatically inactive PLA2G12B is mainly expressed in the liver, small intestine, and kidney [42], PLA2G12A is strongly expressed in the human heart and skeletal muscle, kidney, pancreas [41], and intestinal tissues [43]. PLA2G12 are suggested to mediate their physiological roles in part via alternative mechanisms independent of their catalytic activity [41,42,44], which at present are poorly characterized.

Many studies have suggested different roles for sPLA2s in relation to inflammation and cancer. Their functional roles are incompletely understood and seem to be dependent on the enzyme studied, the tissue, and the cancer type involved [17]. sPLA2s play a pro-tumorigenic role in breast, lung, prostate, ovarian, and esophageal cancers, and on the contrary, play an anti-tumorigenic role in gastric and intestinal cancers [18]. Alterations in the expression of diverse sPLA2s, such as groups II, III, and X, are well documented in CRC [19]. However, sPLA2s-associated signaling pathways are yet to be elucidated. Schewe et al. provided a potential mechanism by which secreted PLA2G2A, another family member of sPLAs, suppresses colon cancer by inhibiting Wnt signaling through the intracellular activation of Yap1 [45]. On this note, Ganesan et al. also reported an association between PLA2G2A expression and components of the Wnt signaling pathway, including β-catenin and the Wnt target gene EphB2 in gastric cancer [46].

Remarkably, using a gene set enrichment analysis, we found that *PLA2G12A* expression correlated with EMT signaling pathways, indicating a potential role for PLA2G12A in the initiation of metastasis. In fact, the in vitro data generated in this study indicate that the downregulation of *PLA2G12A* increases malignant behavior in intestinal cells, altering tumor cell growth, migration, and invasion. EMT induction could explain this malignant behavior found in our cell lines when *PLA2G12A* is inhibited. In this regard, different works demonstrate an association between phospholipases and transforming growth factor β (TGF-β)-induced EMT in hepatic cells [47], as well as in breast cancer [48]. In other studies on breast cancer, it has also been found that arachidonic acid, the main product of phospholipase activity, promotes migration and invasion through a PI3K/Akt-dependent pathway [49,50]. However, all these associations still need to be demonstrated and validated in colon cancer cells.

## 4. Materials and Methods

### 4.1. Fly Strains

*Drosophila melanogaster* strains were raised at 25 °C on standard cornmeal media. We used the GAL4/Upstream Activating Sequence (UAS) system [51] for the tissue-specific expression of transgenes. yw hsp70-flp; esg Gal4 UAS-GFP UAS-Ras^V12^/CyO; and UAS-Luciferase FRT82B Gal80/TM6b flies were crossed with yw hsp70-flp; Sp/CyO; and FRT82B Apc2N175KApcQ8/TM6b flies to generate non-metastatic Apc-Ras control clones as described in [26] or to different yw hsp70-flp; UAS-RNAi line/CyO; and FRT82B Apc2N175KApcQ8/TM6b for each gene tested. RNAi lines were obtained from the Bloomington *Drosophila* Stock Center (Indiana, IN, USA) and The Vienna *Drosophila* Resource Center (Vienna, Austria). Apc2N175K is a loss-of-function allele, ApcQ8 is a null allele, UAS-Ras^V12^ is a gain of function transgene, and each gene analyzed had its specific UAS-RNAi line. A mosaic analysis with repressible cell marker (MARCM) [52] clones were generated with a 1 h heat shock at 37 °C of 2–7-day-old females and were marked by the progenitor cell marker escargot (esg) Gal4 line, driving the expression of UAS green fluorescent protein (GFP).

### 4.2. Luciferase Assay

Luciferase assays were performed using the dual-luciferase(R) reporter assay system [27] 2 weeks after MARCM generation. The UAS-luciferase transgene introduced into fly intestine mutant clones confered a linear correlation between the number of mutant cells and the amount of luciferase activity detected. For a hemolymph analysis, hemolymph was extracted from whole flies according to the instructional video published by Laura Musselman (“Drosophila hemolymph collection procedure”. Youtube, uploaded by Laura Musselman, www.YOUTUBE.com/watch?v=im78OIBKlPA 4 November 2013). For whole-fly lysates, flies were squashed using a pipette tip into a luciferase buffer (E1500, Promega, Madison, WI, USA). The samples were loaded into 96-well plates and read on a Tecan Infinite^®^ 200 plate reader (Tecan, Männedorf, Switzerland).

### 4.3. Human Samples

The internal CRC Hospital Universitari Arnau de Vilanova (HUAV) human cohort was composed of 741 patients treated with a surgical resection at the HUAV from January 2010 to December 2020 with pathological stages I–II–III and no residual disease. Patients who received neoadjuvant treatment or had synchronous tumors were excluded from the study. Tumor tissue were collected during surgery and stored with support from the Xarxa de Bancs de Tumors de Catalunya sponsored by the Pla Director d’Oncología de Catalunya (XBTC), IRBLleida Biobank (B.0000682), and Plataforma Biobancos PT20/00021.

Clinical and pathological information of the patients was collected, including age, sex, anatomical location, pathological stage, histology, differentiation degree, venous/lymphatic/perineural invasion, microsatellite status, RAS mutations, adjuvant chemotherapy, RFS, and OS. The OS and TTR durations were defined as the intervals from initial surgery to death and from initial surgery to clinically proven recurrence or metastasis, respectively.

To reduce the size of the cohort and maintain the whole spectrum of clinical outcomes, paired matching was conducted between patients with recurrence (*n* = 90) and without recurrence after more than 5 years of follow up (*n* = 239), accounting for a balanced pathological stage, differentiation degree, and anatomical location. The final cohort was composed of 180 patients (Appendix A).

### 4.4. Real-Time PCR (RT-PCR)

Total RNA was successfully extracted from 153 patients from the internal CRC human HUAV cohort using the Maxwell^®^ RSC simplyRNA Tissue Kit (Promega), and cDNA was synthesized from mRNA using the Reverse Transcriptase assay iScript™ cDNA Synthesis Kit (BioRad, Hercules, CA, USA). RT-PCR was performed using predesigned hydrolysis fluorescent probes for the genes of interest (Integrated DNA Technologies, Coralville, IA, USA) with 40 cycles at 95 °C for 15 s and 60 °C for 1 min using the QuantStudio 7 (ThermoFisher Scientific, Waltham, MA, USA). Relative mRNA expression levels were calculated using the 2^−ΔΔCt^ method, with β-actin as a housekeeping gene control.

### 4.5. TCGA Data Analysis

RNA sequencing data and clinico-pathological information of 592 patients were obtained from Colorectal Adenocarcinoma of The Cancer Genome Atlas (TCGA) and PanCancer Atlas [53] databases. Gene expression data were explored through the Broad Institute Memorial Sloan Kettering Cancer Center cBioPortal and downloaded for an analysis [54,55]. For a gene set enrichment analysis, we used the GenePattern platform from the Gene Set Enrichment Analysis (GSEA) software (UC San Diego and Broad Institute) [56].

### 4.6. Cell Culture Assays

Human CC cell lines used in our experiments, HCT116, HT29, and SW480, were obtained from the American Type Culture Collection (Manassas, VA, USA). Cell lines were cultured in a DMEM (Dulbecco’s modified Eagle’s medium) supplemented with 10% fetal bovine serum, 100 μg/mL of penicillin, and 100 U/mL of streptomycin (all from ThermoFisher Scientific) at 37 °C in a 5% CO_2_ incubator.

### 4.7. Proliferative Assays

Cells were transfected with interference RNA (iRNA) (Integrated DNA Technologies) with Lipofectamine RNAiMAX (ThermoFisher Scientific) against the specific gene or with scramble iRNA for the controls. Gene-specific inhibition was assessed with a RT-PCR and Western blot (Appendix A). The cell growth ability was monitored by seeding 6000–10,000 cells/well of transfected cells in 24-well dishes and counting the cell number every 24 h for 3 days in a Neubauer chamber.

### 4.8. Cell Migration Assays

Migration rates were measured with a scratch wound healing assay where 500,000 transfected cells/well were seeded onto 6-well plates. After 6 h, a wound was scratched using a sterile pipette tip and the cells were washed with PBS (phosphate-buffered saline) until there were no floating cells. Photographs were taken using an optical microscope from 0 to 72 h. Migration rates (mm^2^/hour) were calculated by measuring the decrease in the wound area.

### 4.9. Invasion Assays

In total, 50,000 transfected cells were seeded into the upper chamber of a matrigel-treated transwell (ThermoFisher Scientific), while the bottom well contained a medium supplemented with glucose as an induction stimulus. After 48 h of incubation, cells in the upper chamber that had not migrated through the filter were wiped off with a cotton swab, whereas those that had migrated to the bottom surface were fixed with 4% paraformaldehyde (Sigma-Aldrich, St. Louis, MO, USA) for 15 min, stained with Hoechst (1:400) (ThermoFisher Scientific), and counted under a fluorescence microscope.

### 4.10. Anchorage-Independent Cell Growth Assays

Six-well plates were coated with a 0.50% agar (ThermoFisher Scientific) layer and after solidifying, 6000 cells/well of transfected cells were seeded in a second superimposed layer at 0.35% agar. After incubation for 15 days, when colonies were visible, clones were stained with crystal violet (0.005%) (ThermoFisher Scientific) and the number of colonies was counted under an optical microscope.

### 4.11. Statistical Analyses

Results are presented as the mean ± standard error of the mean (SEM) and statistical differences were determined with a Student’s *t*-test, Fisher’s test, or Mann–Whitney U test as appropriate and corrected with the Benjamini–Hochberg procedure if needed. All statistical tests were two-tailed and *p* values < 0.05 were considered statistically significant. For survival analyses, patients were included in low or high groups if gene levels were respectively below or above the cutoff, defined as the median for each independent marker. A Kaplan–Meier survival analysis was used for time-to-event variables, and the log-rank test was used to determine significance. A Cox proportional hazards model was used to calculate the univariate hazard ratios (HR) ±95% confidence intervals (CI) for OS, TTR, and RFS. Statistical analyses were performed using the Statistical Package for the Social Sciences for Windows (SPSS v.25, Inc., Chicago, IL, USA).

## 5. Conclusions

We propose PLA2G12A as a prognostic biomarker in early-stage CRC, providing evidence that its deficiency promotes tumor growth and dissemination, unveiling novel functions for this poorly characterized phospholipase.

## Figures and Tables

**Figure 1 ijms-24-10889-f001:**
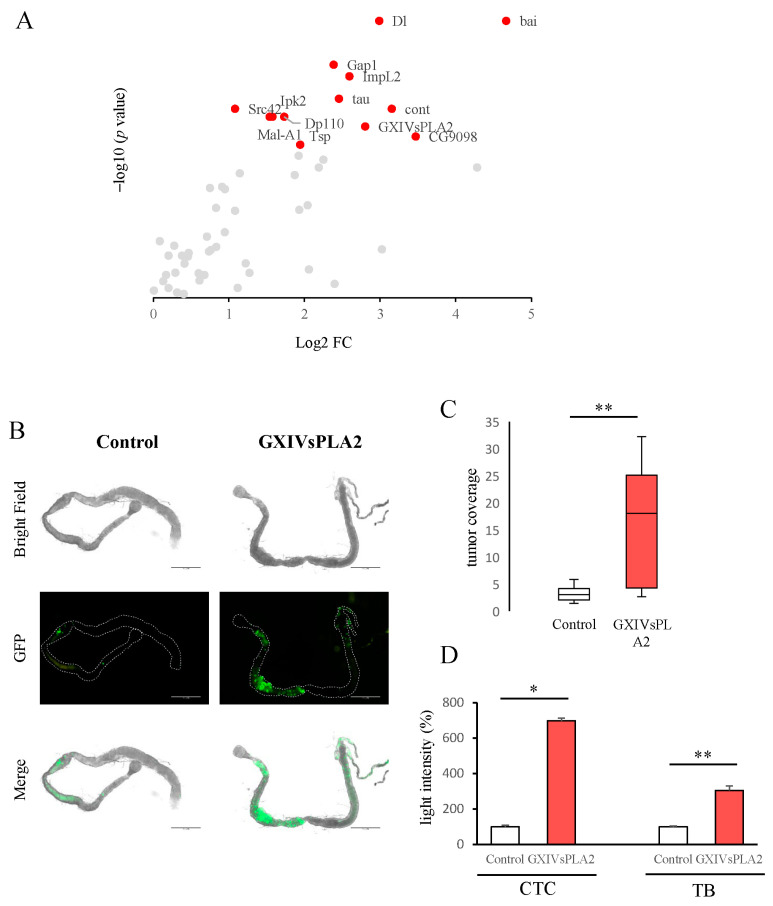
In vivo *Drosophila melanogaster* CRC model results. (**A**) The in vivo genetic screen detected 13 new genes able to modify circulating tumor cell (CTC) number. Data points represent averages of at least 3 replicates of batches of 10 individuals and are represented as the log2 of the fold change (FC) over the control and the –log10 of the *p* value. *p* values were obtained by applying the Mann–Whitney U test and corrected with the Benjamini-Hochberg procedure. Significant genes with *p* < 0.05 are shown in red. (**B**) Representative images of control Apc-Ras and GXIVsPLA2 (orthologous of the human *PLA2G12A*) RNAi *Drosophila* intestines (*n* = 10) 21 days after mutant induction where tumoral cells expressing GFP are visible. Scale bar, 500 µm. (**C**) Quantification of tumor coverage in *Drosophila* intestines represented as the ratio between tumoral GFP cell area and whole intestine area, measured in µm^2^. Box plots show median ± IQR (interquartile range). (**D**) Quantification of CTC number or total tumor burden (TB) in the *Drosophila* CRC model measured in the luciferase assays. Data points represent averages ± SEM shown as percentage with respect to the control Apc-Ras of at least 3 replicates of batches of 10 individuals. * *p* < 0.05; ** *p* < 0.01 by applying the Mann–Whitney U test.

**Figure 2 ijms-24-10889-f002:**
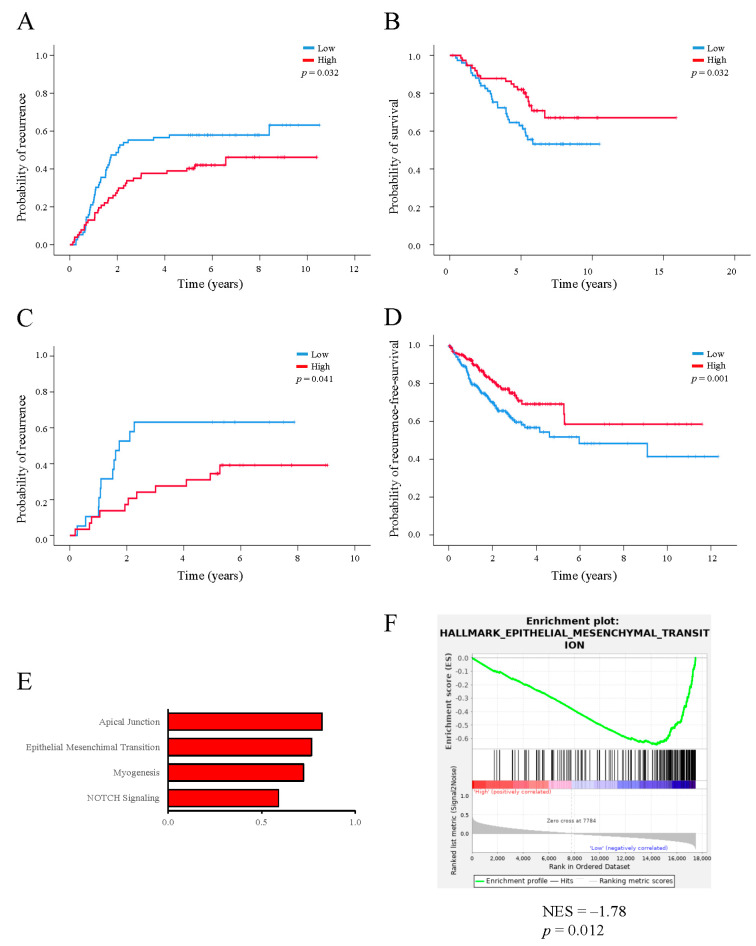
*PLA2G12A* expression in human CRC cohorts is a marker of prognosis. (**A**) Kaplan–Meier survival analysis of the HUAV internal cohort between *PLA2G12A* expression and time to recurrence (TTR), (**B**) *PLA2G12A* expression and overall survival (OS), and (**C**) *PLA2G12A* expression and TTR in stage II patients. (**D**) Kaplan–Meier survival analysis of the TCGA cohort between *PLA2G12A* expression and recurrence-free survival (RFS). *p* values were obtained by applying the log-rank test. (**E**) Gene expression signatures identified with gene set enrichment analysis (GSEA) on the TCGA RNA data cohort. (**F**) Epithelial-to-mesenchymal transition gene expression was activated when *PLA2G12A* levels were low.

**Figure 3 ijms-24-10889-f003:**
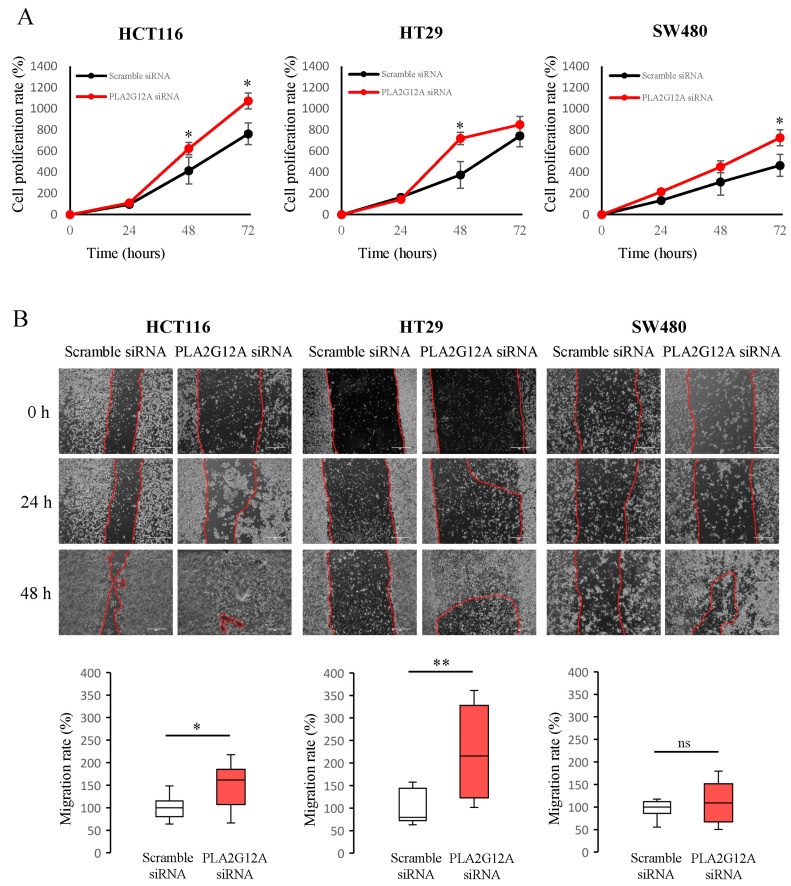
*PLA2G12A* inhibition increases proliferation and migration in CC cells. (**A**) Cell proliferation rates of three different CC cell lines, HCT116, HT29, and SW480 when *PLA2G12A* was downregulated via siRNA. Quantitative values are shown as mean ± SEM of three biological replicates, each of them containing three technical replicates. (**B**) Representative images and quantification of cell migration rates (mm^2^/h) of the scratch wound healing assay in HCT116, HT29, and SW480 cells, respectively. Scale bar, 100 µm. The box plots show median ± IQR of three biological replicates, each of them containing three technical replicates, represented as percentage with respect to the control (scramble siRNA). * *p* < 0.05; ** *p* < 0.01; ns, not significant with Mann–Whitney U test.

**Figure 4 ijms-24-10889-f004:**
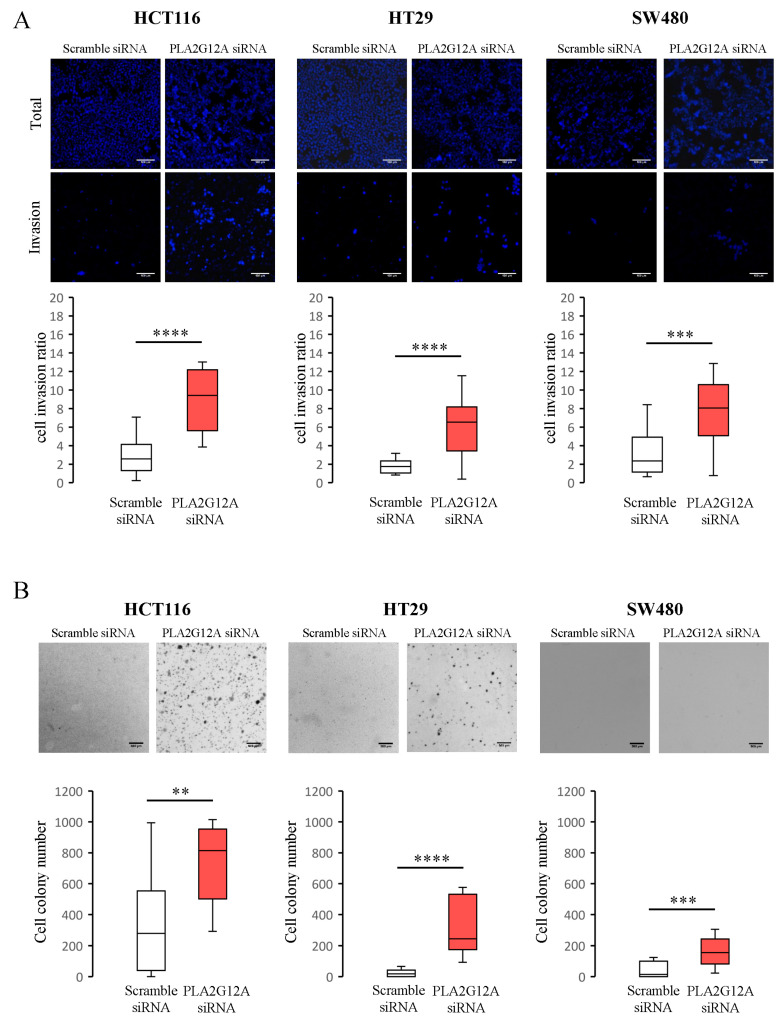
*PLA2G12A* inhibition increases cell invasion and anchorage-independent cell growth. (**A**) Representative images and quantification of cell invasion rates in 3 CC cell lines: HCT116, HT29, and SW480 after *PLA2G12A* downregulation via siRNA. Graphs represent the ratio between the number of cells able to invade the transwell matrigel layer vs. the total cell amount. Scale bar, 100 µm. (**B**) Representative images and quantification of new colonies formed on soft agar plates after *PLA2G12A* downregulation via siRNA in HCT116, HT29, and SW480 cells, respectively. Scale bar, 500 µm. The box plots show median ± IQR of three biological replicates, each of them containing three technical replicates. ** *p* < 0.01; *** *p* < 0.001; **** *p* < 0.0001 by applying the Mann–Whitney U test.

**Table 1 ijms-24-10889-t001:** Hazard ratios (HR) for time to recurrence (TTR) and overall survival (OS).

*Drosophila* Orthologue	Human Gene	TTR			OS		
		HR	95% CI	*p*	HR	95% CI	*p*
Cont	*CNTN4*	0.677	0.432–1.060	0.088	0.602	0.342–1.060	0.079
Dl	*DLL1*	1.316	0.840–2.060	0.230	1.597	0.910–2.805	0.103
Src42	*FRK*	0.847	0.542–1.322	0.464	0.961	0.551–1.675	0.887
Ipk2	*IPMK*	1.078	0.690–1.685	0.740	1.044	0.599–1.820	0.879
Dp110	*PIK3CD*	1.088	0.697–1.698	0.711	0.855	0.490–1.493	0.582
GXIVsPLA2	*PLA2G12A*	0.590	0.375–0.928	**0.022**	0.546	0.308–0.967	**0.038**
Gap1	*RASA3*	0.984	0.631–1.534	0.943	1.287	0.738–2.243	0.375
CG9098	*SH2D3C*	0.833	0.533–1.300	0.421	1.122	0.644–1.957	0.684
Mal-A1	*SLC3A1*	1.037	0.665–1.617	0.872	0.929	0.532–1.620	0.794
Bai	*TMED10*	1.225	0.783–1.915	0.375	1.064	0.609–1.859	0.828

RNA expression, high vs. low. HR < 1 indicating low expression, poor prognosis. HR > 1 indicating high expression, poor prognosis. Significant values with *p* < 0.05 are shown in bold.

**Table 2 ijms-24-10889-t002:** Baseline characteristics of the internal CRC HUAV cohort based on *PLA2G12A* expression.

Characteristic		High	Low	*p* Value
**Sex**	Male (%)	45 (58.4)	51 (67.1)	0.317
	Female (%)	32 (41.6)	25 (32.9)	
**Age**	Median (years)	74.8	75.4	0.755
**Anatomic location**	Cecum (%)	7 (9.1)	3 (3.9)	0.507
	Ascending colon (%)	17 (22.1)	24 (31.6)	
	Hepatic flexure (%)	1 (1.3)	2 (2.6)	
	Transverse colon (%)	2 (2.6)	2 (2.6)	
	Splenic flexure (%)	1 (1.3)	3 (3.9)	
	Descending colon (%)	6 (7.8)	3 (3.9)	
	Sigmoid colon (%)	25 (31.5)	28 (36.8)	
	Rectosigmoid junction (%)	1 (1.3)	5 (6.6)	
	Rectum (%)	17 (22.1)	6 (7.9)	
**Stage**	I (%)	4 (5.2)	2 (2.6)	0.063
	II (%)	29 (37.7)	19 (25)	
	III (%)	44 (57.1)	55 (72.4)	
**Histology**	Adenocarcinoma (%)	63 (81.8)	65 (85.5)	0.663
	Mucinous adenocarcinoma (%)	13 (16.9)	11 (14.5)	
	Signet ring cell carcinoma (%)	1 (1.3)	0 (0)	
**Differentiation degree**	Low-grade (%)	65 (84.4)	68 (89.5)	0.473
	High-grade (%)	12 (15.6)	8 (10.5)	
**Venous invasion** [missing 1]	Yes (%)	21 (27.6)	33 (43.4)	0.062
	No (%)	55 (72.4)	43 (56.6)	
**Lymphatic invasion** [missing 3]	Yes (%)	27 (35.5)	34 (45.9)	0.245
	No (%)	49 (64.5)	40 (54.1)	
**Perineural invasion** [missing 3]	Yes (%)	21 (28)	28 (37.3)	0.296
	No (%)	54 (72)	47 (62.7)	
**Microsatellite status** [missing 80]	MSS (%)	39 (95.1)	30 (93.8)	1
	MSI (%)	2 (4.9)	2 (6.3)	
**RAS** [missing 122]	RAS mut (%)	6 (60)	12 (57.1)	1
	RAS wt (%)	4 (40)	9 (42.9)	
**Adjuvant chemotherapy**	Yes (%)	38 (49.4)	47 (61.8)	0.144
	No (%)	39 (50.6)	29 (38.2)	

*p* value based on Fisher test (categorical variables) and Student’s *t*-test (continuous variables). Anatomic location (left vs. right), stage (III vs. others), histology (adenocarcinoma vs. others).

## Data Availability

The data generated in this study are available within the article and its Appendix A. Other expression profile data analyzed in this study were obtained from the Broad Institute Memorial Sloan Kettering Cancer Center cBioPortal and Colorectal Adenocarcinoma TCGA PanCancer Atlas (https://www.cancer.gov/tcga, id=coadread_tcga_pan_can_atlas_2018, accessed on 17 March 2022).

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
