# Peer review of "PLA2G12A as a Novel Biomarker for Colorectal Cancer with Prognostic Relevance"

_ijms, 2023, doi:10.3390/ijms241310889_

Round 1

Reviewer 1 Report

The author used an innovative Drosophila CRC model to screen several genes, searching for candidates that modify the migration capabilities of tumour cells. These results were evaluated at the gene expression level in human CRC cohorts and were functionally validated in vitro. Finally, they tried to prove that the PLA2G12A gene is a prognostic biomarker of early-stage CRC, providing evidence that its deficiency promotes tumour growth and dissemination. This study is lacking and needs significant revisions for publication in MDPI's international journal of molecular sciences. But first, let me offer some general and particular feedback to the authors.

General comments: In this study, we identified 13 new genes that modified circulating tumour cell numbers using a genome-wide genetic screen in a CRC model. Candidate genes were subsequently evaluated at the gene expression level in an internal human CRC cohort of 153 patients and an independent cohort from the TCGA, including 592 patients. Though they want to persuade that PLA2G12A is a prognostic biomarker of early-stage CRC using in-vitro functional studies, the articles still need to address some issues to be published. So, I think the work could be published in this journal with major changes.

Specific comments:       

  1. First of all, the author should fix the title as they couldn't fully prove the research topics.
  2. Introduction: This part is very poor. The author should rewrite this part with appropriate references.
  3. In lines 70-71, the author should remove the following sentence "Interestingly, our results revealed that PLA2G12A is a putative tumour suppressor gene involved in dissemination and is associated with a good prognosis in CRC."
  4. The methodology: To prove the expression level, the author needs to work on some tissue samples and cell lines using Western blot.
  5. Could the author explain how they prove that PLA2G12A is a tumour suppressor in early-stage CRC? Patient data didn't support this.
  6. All figure legends' number is italics. Please fix this.
  7. The discussion part should be rewritten with appropriate results and references.
  8. The author didn't follow the instructions for the journal in the reference section.
  9. The language and style of English must be extensively adjusted.

The language and style of English must be extensively adjusted.

Reviewer 2 Report

Parisi et al. used an innovative Drosophila CRC model to analyze the expression of genes that mediate the migration of cancer cells. The authors found the secreted phospholipase PLA2G12A to be expressed in human CRC samples and that this gene is associated with a good prognosis in human CRCs.

This is quite an interesting study but there are some points that have to be clarified.

In detail, I have the following suggestions to improve the manuscript:

Chapter “Introduction”

1.       Line 49: please explain what sort of molecule CDX2 is.

2.       Line 52: I do not understand why you cite the circulating tumor DNA study. Isn´t it clear that when you can detect tumor DNA in the blood after resection that these patients have a high risk of recurrence?

Chapter “Results”

2.1. Screening of candidate genes involved in tumor cell dissemination using an in vivo CRC model

1.       The authors knocked each of the candidate genes down and used a luciferase reporter to quantify the knockdown. With the luciferase reporter you can measure the amount of cells that have been transduced but not the knockdown. With siRNA you can have a terrific wide range of knockdown as you know.

Please show how good your siRNA knockdown worked by qPCR or Western Blot.

2.2 Identification of PLA2G12A as a potential tumorsuppressor gene associated with prognosis in human CRC cohorts

1.       The phospholipase A2 family has 16 group members that can be further divided into six subfamilies, depending on the physiological functions.
To avoid confusion please describe the PLA2G12A in two or three sentences.

2.       You showed that patients with a low expression of PLA2G12A have a worse prognosis compared with patients with a high expression. These results are quite surprising as phospholipases hydrolyse phospholipids, release arachidonic acid, and lysophosphatidic acid. Arachidonic acid is the substrate for COX1 and COX2 enzymes to synthesise eicosanoids including prostaglandins. PGE2 is a known pro-inflammatory lipid mediator and promotes tumor progression.

Please say one or two sentences to this, for me, astonishing association.

3.       How can a phospholipase induce EMT? Is there anything known concerning the mechanisms?

2.3 Functional characterization of PLA2G12A in CRC preclinical models

1.       I could not find the description of Figure 3 in the text. Please add this.

2.       Please show the level of knockdown in a qPCR analysis or a Western Blot or an immunohistological staining. The level of expression makes a difference within three days of culture.

3.       Do the cell lines express PLA2G12A at a comparable level? To answer this question, I suggest Western Blots quantification.

4.       To substantiate your hypothesis that PLA2G12A is necessary for colorectal cancer cells to keep an epithelial phenotype, the authors should analyse E-cadherin as an epithelial marker and vimentin or fibronectin as mesenchymal markers in knockdown cells and controls.

5.       Please show some RNA data of EMT transcription markers like Snail1, Snail2 or ZEB1.

6.       Is PLA2G12A a secreted phospholipase?

Chapter “Discussion”

1.       Line 219: “These are characterized…”. Please add a reference for that.

2.       Line 225: The authors claim, that all 13 genes increase the number of CTCs but in line 90 they explain that for technical reasons they continue with 10 of these genes.

3.       Line 240: please add what is known about the mechanisms of the divergent effects in different cancers.

The manuscript is quite easy to read.

Round 2

Reviewer 2 Report

Chapter “Results”

2.1. Screening of candidate genes involved in tumor cell dissemination using an in vivo CRC model

1. The authors knocked each of the candidate genes down and used a luciferase reporter to quantify the knockdown. With the luciferase reporter you can measure the amount of cells that have been transduced but not the knockdown. With siRNA you can have a terrific wide range of knockdown as you know.

Please show how good your siRNA knockdown worked by qPCR or Western Blot.

This is a very interesting comment. To address this, say that the in vivo model used to analyze circulating tumor cells by luciferase measurements were obtained with stable RNAi Drosophila melanogaster lines from the Bloomington Drosophila Stock Center (Indiana, IN, USA) and The Vienna Drosophila Resource Center (Vienna, Austria).  These companies provide links about their validation methods and displays information about most RNAi lines: https://bdsc.indiana.edu/stocks/rnai/rnai_all.html and https://shop.vbc.ac.at/vdrc_store/

I am terribly sorry but it is not the task of a reviewer or a reader to look up how good the specific siRNA worked. The authors still have to provide data for every gene they knocked down. As you know, siRNA is to knockdown genes and not to knockout. Some siRNAs knockdown the gene to only 80% remaining RNA or protein and some to 20%. If the company you bought the siRNA provides these data, please add this in your manuscript.

2. Please show the level of knockdown in a qPCR analysis or a Western Blot or an immunohistological staining. The level of expression makes a difference within three days of culture.

Thanks for the suggestion. This is a great idea which we had in mind. In order to prove downregulation of PLA2G12A on the in vitro experiments, we added an extra figure to the supplementary material (Supplementary Figure S6) were it is shown the mRNA and protein levels in the three colon cancer cell lines after 48h of PLA2G12A siRNA transfection.

Thank you for adding the Western Blot data. Unfortunately, I could not find the description in the manuscript. As you know, authors have to mention supplementary figures and tables in the main manuscript.

Additionally, please add the time point when you did the analysis e.g. after 24h incubation.

2. Please show the level of knockdown in a qPCR analysis or a Western Blot or an immunohistological staining. The level of expression makes a difference within three days of culture.

Thanks for the suggestion. This is a great idea which we had in mind. In order to prove downregulation of PLA2G12A on the in vitro experiments, we added an extra figure to the supplementary material (Supplementary Figure S6) were it is shown the mRNA and protein levels in the three colon cancer cell lines after 48h of PLA2G12A siRNA transfection.

Thank you for adding the Western Blot data. Unfortunately, I could not find the description in the manuscript. As you know, authors have to mention supplementary figures and tables in the main manuscript.

Additionally, please add the time point when you did the analysis e.g. after 24h incubation.

4. To substantiate your hypothesis that PLA2G12A is necessary for colorectal cancer cells to keep an epithelial phenotype, the authors should analyse E-cadherin as an epithelial marker and vimentin or fibronectin as mesenchymal markers in knockdown cells and controls.

5.       Please show some RNA data of EMT transcription markers like Snail1, Snail2 or ZEB1.  

As it has been suggested in point 4 and 5, we performed a RT-PCR assay of several epithelial and mesenchymal markers, as well as EMT transcription markers. We show the results obtained by the 2-∆∆Ct method in the following figure:

As it can be seen in the graph, mRNA levels of the different markers have a particular behaviour depending on the cell line. Moreover, the variability among experiments is also very high.  We presume that these differences could be explained because the RNAi method used depends on transfection efficiency and is not a permanent method of inhibition. Currently, we are working with shRNA and lentiviral infection in order to obtain cellular lines with a PLA2G12A inhibition maintained over time, which will allow us to analyse cell behaviour for longer periods and obtain more reliable results.

The qPCR data are quite astonishing for me. With your data you show that HCT116 and SW480 upregulate the expression of the epithelial gene E-cadherin after knockdown, although you think that PLA2G12A is necessary for an epithelial phenotype.

When did you harvest the transfected cells and isolated the mRNA? In Figure 3 you show that, although HT29 shows a significantly reduced proliferation rate compared to the controls after 48h, this diminishes after 72h. In the case of HT29 your siRNA seem to work very transiently.

There are two possibilities for these astonishing results. Your methods are not sensitive enough to prove your hypothesis or your hypothesis is wrong.

Round 3

Reviewer 2 Report

In my opinion, the manuscript has very much improoved.

The authors now show the knockdown of the siRNAs and the qPCR of several epithelial and mesenchymal markers after knockdown of PLA2G12A.

And in the discussion paragraph, the authors now do not claim more than the data represent.